



# Cosmogenic nuclide exposure age scatter in McMurdo Sound, Antarctica records Pleistocene glacial history and processes

Andrew J. Christ[1], Paul R. Bierman[1,2], Jennifer L. Lamp[3], Joerg M. Schaefer[3], Gisela Winckler[3]

[1]Gund Institute for Environment, University of Vermont, Burlington, VT, 05405, USA

[2]Rubenstein School of the Environment and Natural Resources, University of Vermont, Burlington, VT, 05405, USA

[3]Lamont Doherty Earth Observatory, Columbia University, Palisades, NY 10964, USA

*Correspondence to*: Andrew Christ (Andrew.Christ@uvm.edu)

**Abstract.** The preservation of cosmogenic nuclides that accumulated during periods of prior exposure but were not subsequently removed by erosion or radioactive decay, complicates interpretation of exposure, erosion, and burial ages used

for a variety of geomorphological applications. In glacial settings, cold-based, non-erosive glacier ice may fail to remove inventories of inherited nuclides in glacially transported material. As a result, individual exposure ages can vary widely across a single landform (e.g. moraine) and exceed the expected or true depositional age. The surface processes that contribute to inheritance remain poorly understood, thus limiting interpretations of cosmogenic nuclide datasets in glacial environments. Here, we present a compilation of new and previously published exposure ages of multiple lithologies in local

Last Glacial Maximum (LGM) and older Pleistocene glacial sediments in McMurdo Sound, Antarctica. Unlike most Antarctic exposure chronologies, we are able to compare exposure ages of local LGM sediments directly against an independent radiocarbon chronology of fossil algae from the same sedimentary unit that brackets the age of the local LGM between 12.3 and 19.6 ka. Cosmogenic exposure ages vary by lithology, suggesting that bedrock source and surface processes prior to, during, and after glacial entrainment explain scatter. [10]Be exposure ages of quartz in granite, sourced from

the base of the stratigraphic section in the Transantarctic Mountains, are scattered but young, suggesting that clasts entrained by sub-glacial plucking can generate reasonable apparent exposure ages. [3]He exposure ages of pyroxene in Ferrar Dolerite, which outcrops above outlet glaciers in the Transantarctic Mountains, are older, which suggests that clasts initially exposed on cliff faces and glacially entrained by rock fall carry inherited nuclides. [3]He exposure ages of olivine in basalt from local volcanic bedrock in McMurdo Sound contain many excessively old ages, but also have a bimodal distribution with peak

probabilities that slightly pre-date and post-date the local LGM; this suggests that glacial clasts from local bedrock record local landscape exposure. With the magnitude and geological processes contributing to age scatter in mind, we examine exposure ages of older glacial deposits and suggest that the most extensive Pleistocene ice sheet inundated McMurdo Sound during Marine Isotope Stage 8. These results underscore how surface processes operating in the Transantarctic Mountains are expressed in the cosmogenic nuclide inventories held in Antarctic glacial sediments.



## 1. Introduction: the problem of inherited cosmogenic nuclides

*In situ* cosmogenic nuclides, which accumulate in near-surface materials during exposure to cosmic radiation, can be measured across a wide range of environments and timescales to quantitatively describe earth surface processes, including quantifying erosion rates (Portenga et al., 2019), dating landforms and surfaces (Christ et al., 2021a; Wells et al., 1995), reconstructing changes in climate over millions of years (Bierman et al., 2016; Schaefer et al., 2016; Shakun et al., 2018), among many other applications. However, inherited nuclides, those which accumulated in surface material during periods of prior exposure but were not subsequently removed by erosion or through the decay of radioactive nuclides, can produce results that are difficult-to-interpret in surface exposure dating samples. Nuclide inheritance affects nearly all geomorphologic settings – it complicates the interpretation of alluvial fan ages in deserts (Owen et al., 2011), river terrace exposure ages used to determine canyon incision rates (Cook et al., 2009), shore platform erosion rates in coastal settings (Trenhaile, 2018) and glacial chronologies using exposure ages of boulders on moraines or glacially sculpted bedrock (Briner et al., 2014; Corbett et al., 2016; Davis et al., 1999; Young et al., 2016).

Inherited nuclides are prevalent in glacial sediments, from alpine glacier moraines (Briner et al., 2005; Heyman et al., 2011) to glacial erratics transported by polar ice sheets (Brook et al., 1995; Corbett et al., 2019; Hein et al., 2014). Surface exposure ages of glacial deposits are assumed to record the duration of exposure following emplacement and/or glacial retreat, allowing a reconstruction of former ice extent at specific time-slices (Dunai, 2010). In practice, exposure chronologies can yield highly scattered exposure ages that vary widely across a single, contemporaneously deposited landform (i.e., a moraine) and often greatly exceed the expected or true depositional age (Hein et al., 2014; Heyman et al., 2011). Cosmogenic nuclide scatter in glacial environments is likely due to clasts or surfaces that contain nuclides that accumulated during periods of prior exposure but were not subsequently removed by non-erosive, cold-based glacial ice (Briner et al., 2006; Corbett et al., 2019; Koester et al., 2020). Cold-based glaciation occurs where ice thickness and/or basal ice temperatures are insufficient to produce meltwater and basal sliding, both of which facilitate subglacial erosion, and therefore removal of inherited nuclides, in formerly exposed surfaces (Bennett and Glasser, 1996). The basal thermal regime below ice can be highly heterogeneous, resulting in selective erosion of the landscape (Briner et al., 2006; Jamieson et al., 2010; Sugden et al., 2005). Consequently, glacial sediments can be entrained, transported, and deposited by both warm- and cold-based glacial ice, and ultimately produce scattered cosmogenic surface exposure ages (Hein et al., 2014; Heyman et al., 2011).

The problem of inherited nuclides and exposure age scatter is exacerbated in Antarctica, where large polythermal ice sheets, hyper-arid polar desert climate conditions, and extremely low sub-aerial erosion rates have persisted since the Miocene (Lamp et al., 2017; Lewis and Ashworth, 2016; Lewis et al., 2008; Shakun et al., 2018; Sugden and Denton, 2004; Sugden et al., 2006). Glacial sediments in ice-free areas of Antarctica preserve direct geological constraints on former ice sheet configurations from Holocene to Miocene time (Anderson et al., 2020; Balco et al., 2013; Balter-Kennedy et al., 2020; Bromley et al., 2010; Christ and Bierman, 2020; Hall et al., 2015; Jones et al., 2015, 2021; Nichols et al., 2019; Spector et



al., 2017). Cosmogenic nuclides are often the only method available for constraining a numerical chronology of Antarctic glacial deposits. Radiocarbon dating is usually not applicable due to the absence of organic remains and/or because the

sedimentary archives are >50 kyr. Low erosion rates have resulted in long-lived exposure of high elevation surfaces that are later covered and/or entrained by cold-based glacier ice that fails to erode and remove inherited nuclides (Balco et al., 2019; Hein et al., 2014; Stone et al., 2003; Sugden et al., 2014). Additionally, debris entrainment via rockfall may introduce nuclide scatter due to differential erosion on cliff-faces (Lamp et al., 2017; Mackay et al., 2014). However, large outlet glaciers and ice streams have also deeply eroded and modified parts of the landscape (Lewis and Ashworth, 2016; Sugden et

al., 2005) and transported sediments to terrestrial margins (Stuiver et al., 1981) and onto the continental shelf (Anderson, 1999; Bentley et al., 2014). As a result, many cosmogenic nuclide exposure chronologies from Antarctic terrestrial glacial deposits contain scatter with exposure ages that can span tens to hundreds of thousands of years for individual landforms (Brook et al., 1993; Hein et al., 2014; Joy et al., 2014; Staiger et al., 2006; Swanger et al., 2011).

## 2. McMurdo Sound: a unique setting to examine cosmogenic nuclide inheritance

75       Here, we focus on cosmogenic nuclide analyses from McMurdo Sound, Antarctica (78°S, 165°E), as it is uniquely suited to examine scatter in exposure age datasets from glacial sediments (Fig. 1A). McMurdo Sound hosts one of the largest ice-free areas in Antarctica and contains Pleistocene glacial sediments on volcanic islands and coastal valleys of the Transantarctic Mountains (Christ and Bierman, 2020; Denton and Marchant, 2000; Hall et al., 2015; Hall and Denton, 2000; Jackson et al., 2018; Stuiver et al., 1981) (Fig. 1, 2). The prominence of the Royal Society Range prevents the East Antarctic

Ice Sheet (EAIS) from directly discharging into McMurdo Sound. During past glacial periods when southerly EAIS outlet glaciers expanded into the Ross Sea, grounded ice circumvented volcanic features and overflowed into McMurdo Sound from the east (Christ and Bierman, 2020; Denton and Marchant, 2000; Greenwood et al., 2018; Hall et al., 2015; Stuiver et al., 1981) (Fig. 1). These incursions of grounded ice into McMurdo Sound transported and deposited lithologies from the Transantarctic Mountains, including Granite Harbor Intrusives, Ferrar Dolerite, Koettlitz Group metamorphic rocks, Beacon

Sandstone, as well as other rocks from the East Antarctic interior (Christ and Bierman, 2020; Denton and Marchant, 2000; Talarico et al., 2012, 2013) (Fig. 1B). Specifically, lithologies present in the AND-1B drill core and erratics in McMurdo Sound correspond primarily to rocks found along the Mullock and Skelton Glaciers (Talarico et al., 2012, 2013). The diversity of lithologies in glacial sediments in McMurdo Sound provides targets for multiple cosmogenic nuclides to calculate exposure ages; yet, previous efforts yielded a complex exposure history (Anderson et al., 2017; Brook et al., 1995;

Joy et al., 2017).



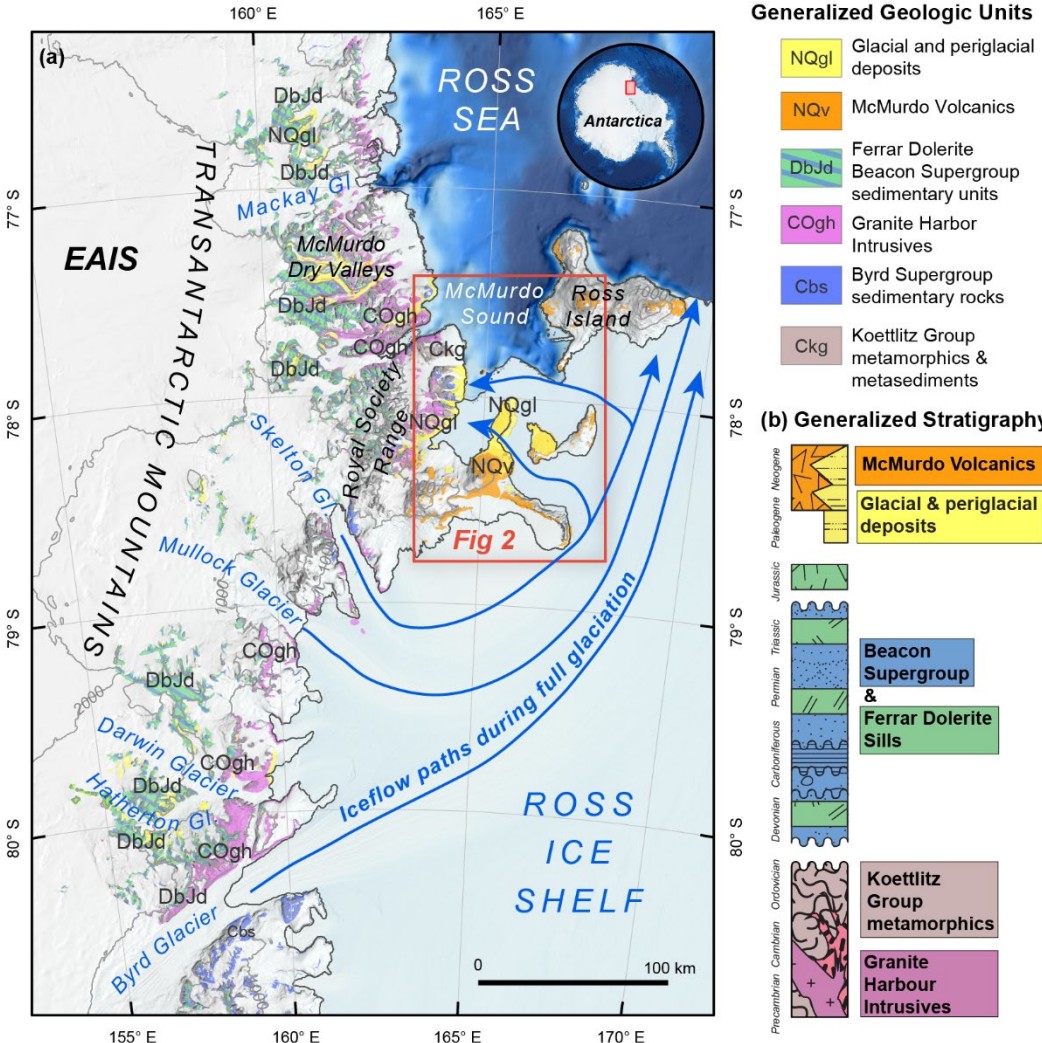

**Figure 1: Bedrock geology and generalized stratigraphy of Southern Victoria Land, Antarctica.** (a) Regional overview map showing the generalized geologic units of Southern Victoria Land, Antarctica. Blue arrows show the general flow directions of outlet glaciers when expanded into the Ross Sea. Base map uses data from the 2013 International Bathymetric Chart of Southern Ocean (Arndt et al., 2013) and the Reference Elevation Mosaic of Antarctica generated by the Polar Geospatial Center (Howat et al., 2019); generalized bedrock geology adapted from (Grindley and Laird, 1969; Warren, 1969).(b) Generalized stratigraphy of Southern Victoria Land, Antarctica adapted from (Fitzgerald, 2002).




McMurdo Sound contains at least two different Pleistocene glacial sedimentary units (Denton and Marchant, 2000). At lower elevations (<510 m), the local Last Glacial Maximum (LGM) glacial deposit, known locally as Ross Sea drift, is characterized by a high concentration of erratic lithologies and un-weathered clasts that lack iron-staining or ventifaction (Christ and Bierman, 2020; Denton and Marchant, 2000; Hall et al., 2015; Stuiver et al., 1981). Unlike many Antarctic

glacial sediments, Ross Sea drift has a robust radiocarbon chronology of fossil algae embedded in moraines and ice-marginal sediments, which indicates that grounded ice maintained its local LGM extent between 12.3 and 19.6 calibrated thousands of years before present (cal. ka) (Christ and Bierman, 2020; Hall et al., 2015; Hall and Denton, 2000; Jackson et al., 2018; Stuiver et al., 1981). A pioneering application of cosmogenic $^{10}$Be, $^{26}$Al, and $^{3}$He in several lithologies from Ross Sea drift produced a highly scattered exposure age dataset ranging from 8 to 106 kyr (Brook et al., 1995). A recent elevation transect

of $^{10}$Be in quartz ($^{10}$Be$_{quartz}$) exposure ages of granite boulders along eastern Mount Discovery indicates that ice thinned from its maximum extent no earlier than 14.1 ka to present conditions by 7.6 ka (Anderson et al., 2017). At higher elevations in McMurdo Sound, other glacial sediments remain largely unmapped and undated. These glacial deposits are characterized by a lower concentration of erratic lithologies relative to Ross Sea drift and are presumably older, as indicated by enhanced surface weathering characteristics including chemically-altered surfaces, and ventifacted and cavernously weathered

boulders (Denton and Marchant, 2000). Earlier reconnaissance mapping on Mount Discovery indicated these deposits extend up to 770 m elevation (Denton and Marchant, 2000; Stuiver et al., 1981). Other similar higher elevation glacial deposits exist above Ross Sea drift in the Royal Society Range and, while they have not been mapped in detail, have some exposure age constraints that range widely between 104 – 567 kyr (Brook et al., 1995).

There are several advantages to examining exposure ages of glacial deposits in McMurdo Sound. Exposure ages of

clasts from Ross Sea drift can be compared against the independently constrained radiocarbon chronology of fossil algae embedded in moraines and ice-marginal sediments from the same deposit (Fig. 2, 3) (Christ and Bierman, 2020; Hall et al., 2015; Hall and Denton, 2000; Jackson et al., 2018; Stuiver et al., 1981). The diversity of lithologies in glacial sediments in McMurdo Sound provides targets for multiple cosmogenic nuclides by which to calculate exposure ages and the different lithologies in glacial sediments from McMurdo Sound can be tied to their bedrock sources in the Transantarctic Mountains

(Talarico et al., 2012, 2013). By examining a compilation of new and previously published (Anderson et al., 2017; Brook et al., 1995) exposure ages of different lithologies in Ross Sea drift, we are able to quantify the magnitude of exposure age scatter and investigate surface processes that contribute to, prevent, or reduce nuclide inheritance in Antarctic terrestrial glacial sediments. With the magnitude and pattern of exposure age scatter known, we then examine previously unmapped and undated glacial deposits that delineate the maximum extent of grounded ice on Mount Discovery.



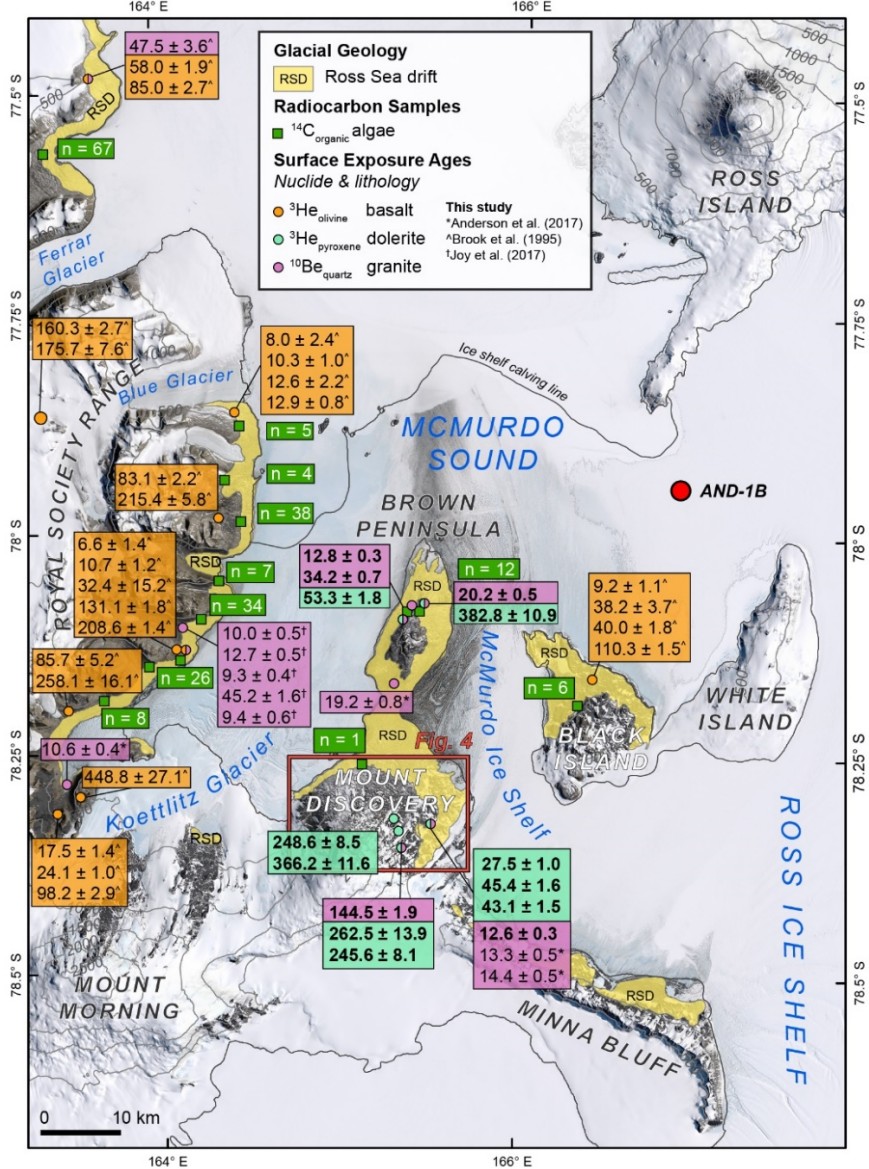

**Figure 2: Compiled cosmogenic nuclide exposure ages and radiocarbon sample locations in McMurdo Sound.** The extent of Ross Sea drift from is shown. Number of calibrated radiocarbon ages shown in green boxes (Christ and Bierman, 2020; Hall et al., 2015; Hall and Denton, 2000; Jackson et al., 2018). All exposure ages from this study and previously published data (Anderson et al., 2017; Brook et al., 1995; Joy et al., 2017) were recalculated using the same LSDn scaling scheme (Lifton et al., 2014) and shown here as the exposure age ± 1σ internal uncertainty from previous studies. Extent of Figure 4 shown in white for detailed mapping on Mount Discovery. AND-1B drill core (red circle) shown for reference (McKay et al., 2012). Base map: Landsat Image Mosaic of Antarctica and 500 m contours derived from DEM(s) created by the Polar Geospatial Center from DigitalGlobe, Inc. imagery (Howat et al., 2019).

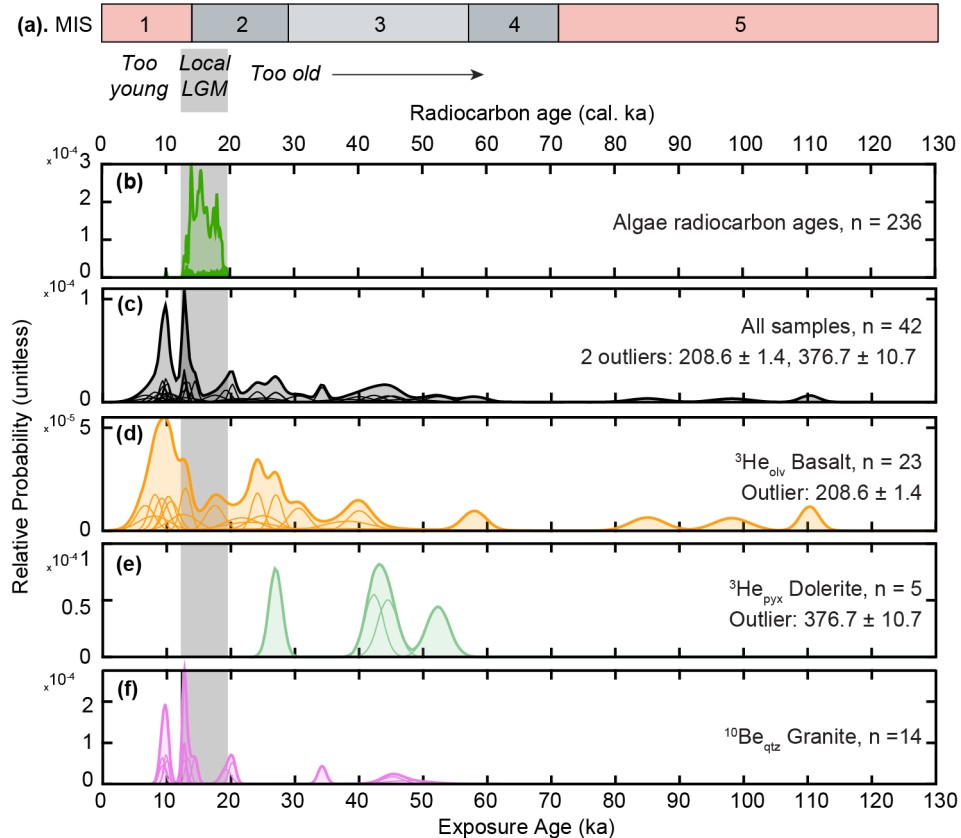

**Figure 3: Ross Sea drift age constraints.** (a) Marine Isotope Stage (MIS) boundaries and the timing of the local Last Glacial Maximum (LGM) in McMurdo Sound (gray box). Kernel density functions of (b) algae radiocarbon ages, and exposure ages in (c) all samples, (d) $^3He_{olv}$ in basalt (e) $^3He_{pyx}$ in dolerite, and (f) $^{10}Be_{qtz}$ in granite. Thin lines show gaussian distributions of individual sample ages and thick lines show the kernel density functions. Outliers are ages >130 ka.

## 3. Materials and methods

### 3.1 Glacial geologic mapping

We built upon the detailed mapping and descriptions of Ross Sea drift on the volcanic islands and peninsulas (Christ and Bierman, 2020; Denton and Marchant, 2000) with new surveys of higher elevation glacial deposits on Mount Discovery (hereafter referred to as the Upper Discovery deposit). The Upper Discovery deposit was mapped at several sites on Mount Discovery at elevations between 450 and 775 m. GPS data points were collected using a handheld Garmin GPS unit (precision <10 m) on moraines and glacial limits. We complemented on-the-ground field mapping with analysis of aerial



photographs, high-resolution (< 1.85 m/pixel) DigitalGlobe, Inc. imagery, and a stereo-photogrammetric digital elevation model (2 m/pixel) created by the Polar Geospatial Center (Howat et al., 2019).

## 3.2 Sample collection

Erratic clasts of Granite Harbor Intrusives (granite) and Ferrar Dolerite were collected for cosmogenic $^{10}$Be in quartz ($^{10}$Be$_{qtz}$)

and $^{3}$He in pyroxene ($^{3}$He$_{pyx}$) exposure dating, respectively, from moraine crests and the limits of Ross Sea drift and the Upper Discovery deposit on the volcanic islands of McMurdo Sound. Clasts that were clearly perched, either on top of other clasts or on the surface of glacial sediment, were collected over those embedded into sediment to avoid 'young' exposure ages due to possible exhumation. Samples were collected from local topographic highs (i.e. moraine ridge crests) to avoid local shielding, clast overturning, rotation, exhumation, or downslope movement. The GPS coordinates, topographic

shielding, clast orientation, clast dimensions, weathering characteristics, and geomorphic setting were recorded for each sample. Samples from larger clasts were collected from the upper 10 cm of rock using a hammer-drill and wedges and shims. Whole-rock samples of smaller cobble-sized clasts were taken.

## 3.3 Cosmogenic nuclide analyses

The upper ~5 cm of each sample was cut using a rock saw at Boston University. Samples were shipped to the Cosmogenic

Nuclide Dating Group at the Lamont-Doherty Earth Observatory for further sample processing and *in situ* nuclide extraction.

### 3.3.1 Cosmogenic $^{3}$He in pyroxene

Samples were crushed to 63-125 μm to separate pyroxene mineral grains, rinsed thoroughly with deionized (DI) water to remove fines, then leached in a 10% phosphoric acid solution overnight to remove oxidation and other contaminants from mineral surfaces. The samples were then density-separated in a centrifuge using sodium polytungstate heavy liquid at a

density of 3.0 g·cm$^{-3}$; the sinking fraction was rinsed with de-ionized water and dried overnight. Magnetic minerals were removed using a Frantz magnetic separator set between 0.4 and 0.5 amps. The non-magnetic fraction was then leached in a 2% HNO$_3$/2% HF solution on a shaker table overnight, then rinsed, dried, and packed in aluminium foil for analysis. Helium isotopes were measured with the LDEO MAP 215–50 noble gas mass spectrometer using an air standard. $^{3}$He in pyroxene ($^{3}$He$_{pyx}$) exposure ages were calculated Version 3 of the online exposure age calculator hosted by the University of

Washington (https://hess.ess.washington.edu/) (Balco et al., 2008) assuming density of 2.9 g·cm$^{-3}$, corrected for shielding and thickness, and used the LSDn scaling scheme (Lifton et al., 2014).

### 3.3.2 Cosmogenic $^{10}$Be in quartz

Samples were crushed to 500-125 μm, then froth flotation was used to separate the quartz grains. Quartz separates were purified by leaching in a HNO$_3$/HF acid solution, and then $^{10}$Be$_{qtz}$ was extracted from purified quartz, both according to



standard protocols in the Cosmogenic Nuclide Dating Group. $^{10}$Be/$^9$Be ratios were measured at Lawrence Livermore
National Laboratory. $^{10}$Be$_{qtz}$ exposure ages were calculated using the version 3 of the online exposure age calculator hosted
by the University of Washington (https://hess.ess.washington.edu/) (Balco et al., 2008) assuming a rock density of 2.7 g cm$^{-3}$
corrected for shielding and thickness, and the LSDn scaling scheme (Lifton et al., 2014). Four procedural blanks were
processed with samples to correct background $^{10}$Be/$^9$Be (Table 3).

**3.4 Compiled exposure chronology**

New and previously published cosmogenic nuclide concentrations from previous studies in McMurdo Sound
(Anderson et al., 2017; Brook et al., 1995; Joy et al., 2017) were compiled from the ICE-D database (antarctica.ice-d.org)
(Balco, 2020). Updated exposure ages for all samples were calculated version 3 of the online exposure age calculator hosted
by the University of Washington (https://hess.ess.washington.edu/) using the LSDn scaling scheme (Lifton et al., 2014).
Only samples collected from the maximum extent of Ross Sea drift were selected from previous studies; we do not include
exposure ages collected from elevation transects and used for deglacial thinning chronologies. All exposure ages of older
glacial deposits, however, were included as geomorphologic mapping of these deposits is less complete. Only $^{10}$Be$_{qtz}$ in
granite, $^3$He in olivine ($^3$He$_{olv}$) in basalt, and $^3$He$_{pyx}$ in dolerite are considered here; too few exposure ages in other erratic
lithologies (such as gneiss and sandstone) and $^{26}$Al$_{qtz}$ exist to be compared against the larger dataset. Additionally, we
exclude previously published $^3$He$_{qtz}$ exposure ages due to diffusion uncertainties inherent to that specific nuclide system
(Brook et al., 1995).

**4. Results**

**4.1 Glacial geologic mapping**

The Upper Discovery deposit reaches a maximum elevation of 770 m and decreases in elevation from east to west
along Mount Discovery (Fig. 4). The limit of the Upper Discovery deposit is nearly continuous and well-defined across the
northern face of Mount Discovery, and is draped onto cinder cones, volcanic vents, and intervening valleys. Windswept
snow fields and alpine glaciers presently cover and cross over the glacial limit in a few locations. The elevation of the glacial
limit descends to ~450 m elevation on western Mount Discovery above Koettlitz Glacier where it becomes poorly defined
due to disturbance from solifluction. Between the limits of the Upper Discovery deposit and Ross Sea drift on Mount
Discovery, no clear moraines or intermediate glacial limits were found. Above the Upper Discovery deposit, no erratic
lithologies are present – only local volcanic rocks exist.



**Figure 4: Upper Discovery deposit mapping and exposure ages.** (a-d) Field photos show the well-defined (solid orange)
and diffuse (dashed orange) glacial limits on upper Mount Discovery. (e) Glacial geologic map and cosmogenic nuclide
exposure ages of glacial deposits on Mount Discovery. Camera icon shows vantage point of field photos in panels (a-d).
Samples with "JA2016-" prefix are from Anderson et al., 2017; all other samples are new age constraints from this study.





The Upper Discovery deposit is characterized by low-relief moraines (< 1 m height) where clasts are mostly

embedded in the soil matrix. Large boulders (>1 m diameter) are rare and most clasts are cobble-sized. Observed erratic

lithologies include Ferrar Dolerite and Granite Harbor Intrusives. Clasts are characterized by varnished surfaces,

ventifaction, and cavernous weathering. Patterned ground is well developed on the surface, and, unlike Ross Sea drift,

thermokarst topography and features are absent along much of the upper elevation limit. Solifluction lobes are present along

the northwest face of upper Mount Discovery and have disturbed the Upper Discovery deposit limit.

**4.2 Cosmogenic nuclide analyses**

**4.2.1 New $^3$He$_{pyx}$ in Ferrar Dolerite exposure ages**

$^3$He$_{pyx}$ exposure ages of Ferrar Dolerite generally sort according to the stratigraphy of mapped glacial deposits but display

wide scatter within these stratigraphic groupings (Table 1). Four of the five $^3$He$_{pyx}$ exposure ages of Ferrar Dolerite in Ross

Sea drift on eastern Mount Discovery and Brown Peninsula range between $27.5 \pm 1.0$ ka to $53.2 \pm 1.1$ ka, while the

remaining exposure age from a sharp-crested moraine on Brown Peninsula was much older, $382.6 \pm 10.9$ ka. $^3$He$_{pyx}$ exposure

ages (n=4) from the Upper Discovery deposit were old; three ages are between $245.7 \pm 8.1$ ka and $262.4 \pm 13.9$ ka; the other

exposure age was even older ($366.0 \pm 11.6$ ka).

Table 1: $^3$He$_{pyroxene}$ exposure ages from Ferrar Dolerite

| Sample Name | Latitude (dd) | Longitude (dd) | Elevation (m) | Location | Shielding correction | Thickness correction[a] | $^3$He concentration atoms g$^{-1}$ | $^3$He error atoms g$^{-1}$ | Exposure Age[b] (ka) | Internal error[b] (ka) | External error[b] (ka) |
|---|---|---|---|---|---|---|---|---|---|---|---|
| *Ross Sea drift* | | | | | | | | | | | |
| ACX-13-08 | -78.3338 | 165.5146 | 512 | Mount Discovery | 0.9993 | 0.9770 | 7.04E+06 | 2.66E+05 | 27.0 | 1.0 | 3.1 |
| ACX-13-09 | -78.3340 | 165.5140 | 511 | Mount Discovery | 0.9993 | 0.9778 | 1.16E+07 | 4.11E+05 | 44.5 | 1.6 | 5.1 |
| ACX-13-12 | -78.3338 | 165.5146 | 512 | Mount Discovery | 0.9976 | 0.9757 | 1.10E+07 | 3.75E+05 | 42.3 | 1.4 | 4.9 |
| ACX-14-05 | -78.0832 | 165.4644 | 340 | Brown Peninsula | 0.9810 | 0.9905 | 8.24E+07 | 2.34E+06 | 376.7 | 10.7 | 42.8 |
| ACX-14-15 | -78.1002 | 165.3507 | 242 | Brown Peninsula | 0.9989 | 0.9836 | 1.05E+07 | 3.61E+05 | 52.3 | 1.8 | 6.0 |
| | | | | | | | | | | | |
| *Upper Discovery deposit* | | | | | | | | | | | |
| ACX-13-48 | -78.3632 | 165.3468 | 775 | Mount Discovery | 0.9957 | 0.9808 | 8.57E+07 | 4.55E+06 | 255.8 | 13.6 | 31.2 |
| ACX-13-52 | -78.3629 | 165.3466 | 774 | Mount Discovery | 0.9957 | 0.9764 | 7.97E+07 | 2.65E+06 | 239.4 | 8.0 | 27.5 |
| ACX-13-61 | -78.3405 | 165.3227 | 702 | Mount Discovery | 0.9969 | 0.9724 | 1.11E+08 | 3.53E+06 | 357.6 | 11.4 | 40.9 |
| ACX-13-68 | -78.3289 | 165.3173 | 707 | Mount Discovery | 0.9976 | 0.9868 | 7.69E+07 | 2.62E+06 | 242.8 | 8.3 | 28.0 |

**Table 1: $^3$He$_{pyx}$ exposure age from Ferrar Dolerite.** [a]Assumed density for all samples = 2.9 g/cm3. [b]Results calculated

using Cronus Online Calculator v3 with LSDn scaling scheme (Lifton et al., 2014).





### 4.2.2 New $^{10}Be_{qtz}$ in granite exposure ages

$^{10}Be_{qtz}$ exposure ages of granite sort according to stratigraphic age and display less scatter than $^3He_{pyx}$ exposure ages (Table 2). $^{10}Be_{qtz}$ exposure ages of clasts from Ross Sea drift on Mount Discovery and Brown Peninsula had two younger and similar ages (12.6 ± 0.3 ka, 12.8 ± 0.3 ka); and two older ages (20.2 ± 0.6 ka, 34.2 ± 0.7 ka). A single exposure age of a
cavernously weathered boulder in the Upper Discovery deposit yielded an age of 144.5 ± 1.9 ka.

Table 2: $^{10}Be_{quartz}$ exposure ages from granite erratics

| Sample Name | Lat. (dd) | Lon. (dd) | Elev. (m) | Location | Quartz mass (g) [a] | LLNL Cathode # | Mass $^9Be$ added (µg) [b] | Uncorrected $^{10}Be/^9Be$ Ratio[c] | ±1σ[c] | Background corrected[d,e] $^{10}Be/^9Be$ Ratio | ±1σ | [$^{10}Be$] (atoms g[-1]) | ±1σ (atoms g[-1]) | Exposure Age[f] (ka) | Internal error[f] (ka) | External error[f] (ka) |
|---|---|---|---|---|---|---|---|---|---|---|---|---|---|---|---|---|
| *Ross Sea Drift* | | | | | | | | | | | | | | | | |
| ACX-13-18 | -78.3331 | 165.5153 | 507 | Mount Discovery | 10.0477 | BE41642 | 186.8 | 9.17E-14 | 2.09E-15 | 9.12E-14 | 2.10E-15 | 1.13E+05 | 2.60E+03 | 12.6 | 0.3 | 0.8 |
| ACX-14-08 | -78.0813 | 165.4635 | 345 | Brown Peninsula | 4.2853 | BE40972 | 192.2 | 5.21E-14 | 1.38E-15 | 5.16E-14 | 1.39E-15 | 1.55E+05 | 4.15E+03 | 20.2 | 0.5 | 1.3 |
| ACX-14-16 | -78.1007 | 165.3478 | 249 | Brown Peninsula | 7.2061 | BE40973 | 191.7 | 1.34E-13 | 2.54E-15 | 1.33E-13 | 2.55E-15 | 2.37E+05 | 4.53E+03 | 34.2 | 0.7 | 2.1 |
| ACX-14-26 | -78.0882 | 165.3833 | 228 | Brown Peninsula | 10.0188 | BE41643 | 187.0 | 7.07E-14 | 1.56E-15 | 7.02E-14 | 1.57E-15 | 8.75E+04 | 1.96E+03 | 12.8 | 0.3 | 0.8 |
| *Upper Discovery deposit* | | | | | | | | | | | | | | | | |
| ACX-13-56 | -78.3618 | 165.3476 | 772 | Mount Discovery | 10.0216 | BE41644 | 187.2 | 1.28E-12 | 1.63E-14 | 1.28E-12 | 1.63E-14 | 1.60E+06 | 2.03E+04 | 144.5 | 1.9 | 9.0 |

**Table 2: $^{10}Be_{qtz}$ exposure ages from granite erratics.** [a] Shielding correction of 1 applied to all samples. Thickness correction of 0.9833 was applied to all samples assuming a density of 2.7 g cm[-3] for all samples. [b] $^9Be$ was added to samples prepared at LDEO through a beryl carrier made at LDEO with a concentration of 1034.9 µg mL[-1]. [c] Isotopic analysis was
conducted at LLNL Laboratory; ratios were normalized against standard 07KNSTD3110 with an assumed ratio of 2850 x 10-15 (Nishiizumi et al., 2007). [d] A blank correction of 4.6 ± 1.7E-16 (average2016Jun03) was applied to $^{10}Be/^9Be$ measurements from samples ACX-13-18, ACX-14-26, ACX-13-56. See Table 3 for details. [e] A blank correction of 4.9 ± 1.4E-16 (BLK-2016Apr12) was applied to 10Be/9Be measurements from samples ACX-14-08, ACX-14-16. See Table 3 for details. [f] Results calculated using Cronus Online Calculator v3 with LSDn scaling scheme (Lifton et al., 2014).


Table 3: $^{10}Be$ blank correction

| BLANK | Lab ID | Mass $^9Be$ Added ug | $^{10}Be/^9Be$ Ratio | ±1σ | BLANK $^{10}Be$ (atoms/g) | ±1σ (atoms/g) |
|---|---|---|---|---|---|---|
| BLK1-2016Jun03 | BE41648 | 187.4 | 3.5E-16 | 2.0E-16 | 4.34E+03 | 2.46E+03 |
| BLK2-2016Jun03 | BE41649 | 187.4 | 5.8E-16 | 1.5E-16 | 7.20E+03 | 1.88E+03 |
| *Average-2016Jun03* | | | *4.6E-16* | *1.7E-16* | *5.77E+03* | *2.87E+02* |
| BLK-2016Apr12 | BE40971 | 213.0 | 4.9E-16 | 1.4E-16 | 6.96E+03 | 1.98E+03 |

**Table 3: Blank correction for $^{10}Be$ measurements.**



## 4.3 Compiled cosmogenic nuclide ages

### 4.3.1 Ross Sea drift

Exposure ages of Ross Sea drift (n = 42) vary widely, ranging from 6.6 ± 1.4 ka to 376.7 ± 10.7 ka (Table 4). A
probability density function of all exposure ages returns primary and secondary modes at 12.7 ka and 9.8 ka, respectively
(Fig. 3). Although the primary mode falls within the radiocarbon-constrained timeframe of the local LGM (12.3-19.6 ka),
only 9 of the 42 exposure ages from Ross Sea drift fall within this window. Ten exposure ages are <12.3 ka and 23
exposures ages are >19.6 ka. Most of these excessively old ages are within the last glacial period (MIS 2 - 4), but several
exposure ages date to the last interglacial (MIS 5) and two are much older (208.6 ± 1.4 ka [MIS 7], and 376.7 ± 10.7 ka [MIS
8]).

Ross Sea drift exposure ages vary according to lithology and nuclide (Fig. 3). $^3He_{olv}$ in basalt exposure ages ((Brook
et al., 1995), n = 23) vary widely (range: 6.7 – 211 ka), but display two modes, 9.5 ka and 24.2 ka, before and after the local
LGM, respectively. Only three $^3He_{olv}$ in basalt exposure ages are within the local LGM timeframe. $^3He_{pyx}$ in dolerite
exposure ages (n = 5) are consistently older (range: 27.0 ± 1.0 to 376.6 ± 10.7 ka); none of these ages fall within the timing
of the local LGM. $^{10}Be_{qtz}$ in granite exposure ages ((Anderson et al., 2017; Brook et al., 1995; Joy et al., 2017; this study), n
= 14) are scattered (range: 9.3 ± 0.5 to 47.5 ± 3.6 ka) and tend to be younger. The primary mode (12.8 ka) coincides with the
latter half of the local LGM, while an additional secondary mode post-dates maximum ice extent (9.7 ka).

### 4.3.2 Higher-elevation glacial sediments

Exposure ages (n=13) from the higher elevation glacial deposits (including the Upper Discovery deposit) are highly
scattered, and all pre-date the onset of the last glacial period (71 ka, MIS 4) (Fig. 5, Table 4). Several ages clusters coincide
with previous glacial periods MIS 6 (n =4, 131.1 ± 1.7 to 175.7 ± 7.6 ka) and the end of MIS 8 (n = 4, 239.4 ± 8.0 to 258.1 ±
16.1 ka), with several ages dating to MIS 10 (357.6 ± 11.3 ka) and MIS 12 (448.8 ± 27.1 ka). Three ages date to earlier
interglacial periods, including two ages during MIS 5 (83.1 ± 2.2 ka, 85.7 ± 5.2 ka) and one age during MIS 7 (215.4 ± 5.8
ka).





TABLE 4: Compiled exposure ages from McMurdo Sound glacial deposits

| Sample name | Publication | Latitude (DD) | Longitude (DD) | Elevation (m asl) | Lithology | Nuclide (target mineral) | Exposure Age (kyr) | ±1σ (kyr) |
|---|---|---|---|---|---|---|---|---|
| *Ross Sea Drift* | | | | | | | | |
| JA2016-D161 | Anderson et al., 2017 | -78.2793 | 163.4934 | 98 | granite | Be-10 (qtz) | 9.9 | 0.4 |
| JA2016-D183 | Anderson et al., 2017 | -78.3335 | 165.5147 | 526 | granite | Be-10 (qtz) | 13.3 | 0.5 |
| JA2016-D186 | Anderson et al., 2017 | -78.3345 | 165.5111 | 514 | granite | Be-10 (qtz) | 14.4 | 0.5 |
| JA2016-D139 | Anderson et al., 2017 | -78.1719 | 165.2958 | 307 | granite | Be-10 (qtz) | 19.2 | 0.8 |
| KBA89-240 | Brook et al., 1995 | -78.1333 | 164.1500 | 253 | basalt | He-3 (ol) | 6.6 | 1.4 |
| BAK90-138 | Brook et al., 1995 | -77.8667 | 164.4167 | 265 | basalt | He-3 (ol) | 8.0 | 2.4 |
| BAK90-139-1 | Brook et al., 1995 | -77.8667 | 164.4167 | 265 | basalt | He-3 (ol) | 8.2 | 1.0 |
| KBA89-295 | Brook et al., 1995 | -78.1667 | 166.4000 | 393 | basalt | He-3 (ol) | 9.2 | 1.1 |
| BAK90-137-1 | Brook et al., 1995 | -77.8667 | 164.4167 | 265 | basalt | He-3 (ol) | 10.3 | 1.0 |
| KBA89-241 | Brook et al., 1995 | -78.1333 | 164.1500 | 253 | basalt | He-3 (ol) | 10.7 | 1.2 |
| BAK90-137-2 | Brook et al., 1995 | -77.8667 | 164.4167 | 265 | basalt | He-3 (ol) | 12.6 | 2.2 |
| BAK90-136 | Brook et al., 1995 | -77.8667 | 164.4167 | 265 | basalt | He-3 (ol) | 12.9 | 0.8 |
| RA92-039 | Brook et al., 1995 | -78.3183 | 163.4169 | 345 | basalt | He-3 (ol) | 17.5 | 1.4 |
| BAK90-139-2 | Brook et al., 1995 | -77.8667 | 164.4167 | 265 | basalt | He-3 (ol) | 21.7 | 2.8 |
| KBA-89-246-2-3 | Brook et al., 1995 | -78.1333 | 164.1500 | 253 | basalt | He-3 (ol) | 22.9 | 4.2 |
| RA92-040 | Brook et al., 1995 | -78.3183 | 163.4169 | 345 | basalt | He-3 (ol) | 24.1 | 0.9 |
| KBA-89-246-2-1 | Brook et al., 1995 | -78.1333 | 164.1500 | 253 | basalt | He-3 (ol) | 25.0 | 2.4 |
| KBA89-247 | Brook et al., 1995 | -78.1333 | 164.1500 | 253 | basalt | He-3 (ol) | 27.0 | 1.0 |
| KBA-89-246-2-2 | Brook et al., 1995 | -78.1333 | 164.1500 | 253 | basalt | He-3 (ol) | 30.5 | 1.6 |
| KBA89-245 | Brook et al., 1995 | -78.1333 | 164.1500 | 253 | basalt | He-3 (ol) | 32.4 | 15.2 |
| KBA89-293 | Brook et al., 1995 | -78.1667 | 166.4000 | 393 | basalt | He-3 (ol) | 38.2 | 3.7 |
| KBA89-296 | Brook et al., 1995 | -78.1667 | 166.4000 | 393 | basalt | He-3 (ol) | 40.0 | 1.8 |
| KBA89-140 | Brook et al., 1995 | -77.4833 | 163.6667 | 309 | basalt | He-3 (ol) | 58.0 | 1.9 |
| KBA89-142 | Brook et al., 1995 | -77.4833 | 163.6667 | 309 | basalt | He-3 (ol) | 85.0 | 2.7 |
| RA92-041 | Brook et al., 1995 | -78.3183 | 163.4169 | 345 | basalt | He-3 (ol) | 98.2 | 2.8 |
| KBA89-292 | Brook et al., 1995 | -78.1667 | 166.4000 | 393 | basalt | He-3 (ol) | 110.3 | 1.5 |
| KBA89-243 | Brook et al., 1995 | -78.1333 | 164.1500 | 253 | basalt | He-3 (ol) | 208.6 | 1.4 |
| KBA89-143 | Brook et al., 1995 | -77.4833 | 163.6667 | 309 | granite | Be-10 (qtz) | 47.5 | 3.6 |
| MV2-4 | Joy et al., 2017 | -78.1120 | 164.1808 | 40 | granite | Be-10 (qtz) | 9.3 | 0.4 |
| MV3-2 | Joy et al., 2017 | -78.0990 | 163.9871 | 120 | granite | Be-10 (qtz) | 9.4 | 0.6 |
| MV2-1 | Joy et al., 2017 | -78.1090 | 164.1308 | 72 | granite | Be-10 (qtz) | 10.0 | 0.5 |
| MV2-3 | Joy et al., 2017 | -78.0990 | 164.1015 | 27 | granite | Be-10 (qtz) | 12.7 | 0.5 |
| MV2-5 | Joy et al., 2017 | -78.1080 | 164.1343 | 75 | granite | Be-10 (qtz) | 45.2 | 1.6 |
| ACX-13-08 | This publication | -78.3338 | 165.5146 | 512 | dolerite | He-3 (pyx) | 27.0 | 1.0 |
| ACX-13-12 | This publication | -78.3338 | 165.5146 | 512 | dolerite | He-3 (pyx) | 42.3 | 1.4 |
| ACX-13-09 | This publication | -78.3340 | 165.5140 | 511 | dolerite | He-3 (pyx) | 44.5 | 1.6 |
| ACX-14-15 | This publication | -78.1002 | 165.3507 | 242 | dolerite | He-3 (pyx) | 52.3 | 1.8 |
| ACX-14-05 | This publication | -78.0832 | 165.4644 | 340 | dolerite | He-3 (pyx) | 376.7 | 10.7 |
| ACX-13-18 | This publication | -78.3331 | 165.5153 | 507 | granite | Be-10 (qtz) | 12.6 | 0.3 |
| ACX-14-26 | This publication | -78.0882 | 165.3833 | 228 | granite | Be-10 (qtz) | 12.8 | 0.3 |
| ACX-14-08 | This publication | -78.0813 | 165.4635 | 345 | granite | Be-10 (qtz) | 20.2 | 0.5 |
| ACX-14-16 | This publication | -78.0982 | 165.3567 | 220 | granite | Be-10 (qtz) | 34.2 | 0.7 |
| | | | | | | | | |
| *Older Deposits* | | | | | | | | |
| BAK90-141-1 | Brook et al., 1995 | -77.8667 | 163.3833 | 495 | basalt | He-3 (ol) | 160.3 | 2.7 |
| BAK90-141-2 | Brook et al., 1995 | -77.8667 | 163.3833 | 495 | basalt | He-3 (ol) | 175.7 | 7.6 |
| BAK90-214 | Brook et al., 1995 | -78.3000 | 163.5500 | 480 | basalt | He-3 (ol) | 448.8 | 27.1 |
| BAK90-247 | Brook et al., 1995 | -78.2000 | 163.4667 | 545 | basalt | He-3 (ol) | 258.1 | 16.1 |
| BAK90-249-2 | Brook et al., 1995 | -78.2000 | 163.4667 | 545 | basalt | He-3 (ol) | 85.7 | 5.2 |
| BAK90-262-1 | Brook et al., 1995 | -77.9833 | 164.3167 | 510 | basalt | He-3 (ol) | 83.1 | 2.2 |
| BAK90-263 | Brook et al., 1995 | -77.9833 | 164.3167 | 510 | basalt | He-3 (ol) | 215.4 | 5.8 |
| KBA89-239 | Brook et al., 1995 | -78.1333 | 164.1167 | 374 | basalt | He-3 (ol) | 131.1 | 1.7 |
| *Upper Discovery Deposit* | | | | | | | | |
| ACX-13-48 | This publication | -78.3632 | 165.3468 | 775 | dolerite | He-3 (pyx) | 255.8 | 13.6 |
| ACX-13-52 | This publication | -78.3629 | 165.3466 | 774 | dolerite | He-3 (pyx) | 239.4 | 8.0 |
| ACX-13-56 | This publication | -78.3618 | 165.3476 | 772 | granite | Be-10 (qtz) | 144.5 | 1.9 |
| ACX-13-61 | This publication | -78.3405 | 165.3227 | 702 | dolerite | He-3 (pyx) | 357.6 | 11.4 |
| ACX-13-68 | This publication | -78.3289 | 165.3173 | 707 | dolerite | He-3 (pyx) | 242.8 | 8.3 |

**Table 4: Compiled exposure ages from McMurdo Sound glacial deposits.** Cosmogenic nuclide data for Anderson et al., 2017; Brook et al., 1995; Joy et al., 2017 accessed using the ICE-D database (Balco, 2020). Exposure ages calculated with version 3 of the online exposure age calculator hosted by the University of Washington (https://hess.ess.washington.edu/) using the LSDn scaling scheme (Lifton et al., 2014). Sample coordinates from (Brook et al., 1995) are approximate.




**Figure 5: Exposure ages of McMurdo Sound glacial deposits in the context of paleoclimate since 500 ka.** (a) Marine Isotope Stage boundaries showing interglacial (light red) and glacial (light blue) periods, and global mean sea level (Spratt and Lisiecki, 2016) - dashed blue lines show minimum global sea level during the LGM and MIS 8. (b) Temperature difference from modern in the EPICA Dome C ice core in East Antarctica (Jouzel et al., 2007). (c) Exposure ages (±1σ error)

of Ross Sea drift (yellow circles) and older pre-LGM deposits (orange circles), including the Upper Discovery deposit (orange squares) plotted versus elevation. Dark blue band across all panels shows proposed late MIS 8 age of the Upper Discovery deposit.




## 5. Discussion

### 5.1 New insights from inherited nuclides and exposure age scatter

In former and active ice sheet environments, cosmogenic nuclides in glacial sediments often do not yield simple, easy-to-interpret exposure ages; instead, cosmogenic nuclides record the integrated effect of many surface processes operating on a landscape through time. Rather than focusing exclusively on the age of ice extent or retreat, cosmogenic nuclides can also be used to disentangle how glacial sediment is entrained from bedrock sources, transported by glacier ice, and deposited on the landscape. By taking advantage of the relatively simple geology of the Transantarctic Mountains, an

independent (albeit rare) radiocarbon chronology, and analysis of multiple nuclides and lithologies in Ross Sea drift, we can better quantify the magnitude of and processes responsible for exposure age scatter. While scatter in exposure ages from Ross Sea drift prevents precise age constraints of maximum ice extent, it is possible to assign ages at glacial-interglacial timescales. With this information, we then consider the age of older Pleistocene glacial deposits in McMurdo Sound.

### 5.2 Scattered exposure ages of Ross Sea drift

      The narrow range of calibrated radiocarbon ages (12.3 – 19.6 cal. ka) and scattered range of exposure ages that date to within the last glacial period confirm that Ross Sea drift is the youngest glacial deposit recording ice sheet expansion into McMurdo Sound. However, the compiled cosmogenic nuclide exposure chronology derived from multiple lithologies is highly scattered, and most individual exposure ages exceed the timeframe of local LGM ice extent established by the

radiocarbon chronology. While many ages are 'too old' and suggest that these clasts carry an inventory of inherited nuclides, most of the exposure ages (with 3 exceptions) are between the early Holocene (Marine Isotope Stage [MIS] 1) and last glacial period (MIS 2 - 4). Upper elevation, weathered deposits yield much older Pleistocene ages (MIS 5 – 12) that are incompatible with a local LGM depositional age.

      This exposure age dataset underscores the difficulty of using surface exposure dating in Antarctica to constrain

precise ages of individual landforms or glacial deposits from the local LGM using $^{10}Be_{qtz}$, $^{3}He_{pyx}$, or $^{3}He_{olv}$. The most extreme example of scatter comes from two exposure ages from the same moraine ridge on northern Brown Peninsula. This sharp-crested moraine ridge is independently dated to 17.9 cal. ka (Christ and Bierman, 2020), but nearly adjacent clasts yield exposure ages that are both 'too old' and widely different. A perched granite cobble produced an exposure age of 20.2 ± 0.5 ka (ACX-14-08) and a nearby dolerite cobble has an exposure age of 382.8 ± 10.9 ka (ACX-14-05), the oldest of any sample

from Ross Sea drift. On eastern Mount Discovery (512 m), the glacial limit is marked by a stark lithological contrast between erratic-rich Ross Sea drift lying at lower elevations than volcanic-rich glacial deposits (Anderson et al., 2017; Christ and Bierman, 2020; Denton and Marchant, 2000). Here, granite exposure ages coincide with the end of the local LGM in McMurdo Sound (12.6 to 14.4 ka, n = 3; (this study; Anderson et al., 2017), but exposure ages of dolerite from the same location are consistently older (27.5 to 45.5 ka, n = 3). Clearly, exposure ages from individual landforms from the same local

LGM deposit in McMurdo Sound contain clasts with different exposure histories.



### 5.3 Surface processes contributing to exposure age scatter

We suggest that scatter is related to bedrock source and that specific surface processes prevent, enhance, or reduce the likelihood of nuclide inheritance prior to and during glacial entrainment and deposition. Exposure age scatter sorts according to lithology and nuclide: granite $^{10}Be_{qtz}$ exposure ages are younger, but scattered, dolerite $^{3}He_{pyx}$ ages are
consistently too old, and basalt $^{3}He_{olv}$ ages slightly pre-date and post-date the local LGM. These patterns can be explained by examining how clasts were glacially entrained and incorporated into Ross Sea drift.

### 5.3.1 Sub-glacial plucking reduces nuclide inheritance

Granite clasts in Ross Sea drift, sourced from Granite Harbor Intrusives, are likely entrained by sub-glacial plucking, and therefore produce younger, less-scattered exposure ages. Granite Harbor Intrusives comprise the base of the
stratigraphic section for much of the Transantarctic Mountains in Southern Victoria Land (Fig. 1). The large EAIS outlet glaciers in the Transantarctic Mountains south of McMurdo Sound – Byrd Glacier, Mullock Glacier, and Skelton Glacier – have all incised deep troughs that cut into Granite Harbor Intrusives (Fig. 1). Ice sheet models simulate that these outlet glaciers thicken and expand into the Ross Sea during glacial periods; they are also wet-based and erode large volumes of sediment (Golledge et al., 2013). Rock surfaces at the bottom of outlet glacier troughs are up to 1000 m below sea level and
could not experience exposure to cosmic rays. In addition, the Pleistocene stability and persistence of EAIS outlet glaciers in the Transantarctic Mountains likely prevented prior exposure of the bedrock troughs (Kaplan et al., 2017). Taken together, this means that Granite Harbor Intrusives plucked from the base of outlet glacier troughs are less likely to contain cosmogenic nuclides inherited from periods of exposure prior to local LGM deglaciation in Ross Sea drift. The resulting exposure ages are more likely to record clast emplacement at maximum ice extent.

### 5.3.2 Cliff exposure, rockfall, and supraglacial transport increases nuclide inheritance

Old exposure ages of several granite and all dolerite clasts in Ross Sea drift may reflect prior cliff exposure and entrainment by rockfall along outlet glacier margins. Granite Harbor Intrusive rocks are exposed in cliff faces above the glacier surface closer to the coast and along outlet glacier margins in the Transantarctic Mountains (Fig. 1). During previous interglacial periods or glacial low-stands when outlet glaciers were thinner (Jones et al., 2017), exposure on granite cliff
faces would have accumulated cosmogenic nuclides prior to later glacial entrainment. The effect of prior cliff exposure is likely more important for exposure ages of Ferrar Dolerite because these rocks form many of the peaks and cliff faces above glacier surfaces in the Transantarctic Mountains at elevations >1000 m (Fig. 1). Prolonged cliff exposure combined with low erosion rates (Mackay et al., 2014) at high elevations (where the cosmogenic nuclide production rate is also higher), will increase cosmogenic nuclide inventories in dolerite more than granite clasts. Additionally, $^{3}He_{pyx}$ is a stable cosmogenic
nuclide that does not decay during burial (Dunai, 2010); this further increases the probability of nuclide inheritance in dolerite clasts.



Despite increased outlet glacier surface elevations during glacial periods, cliff faces may not be subjected to wet-based glacial erosion due to insufficient ice thickness. Selective linear erosion below polythermal ice sheets can lead to wet-based, erosive glacial conditions in outlet glacier troughs, but cold-based, non-erosive glacial conditions along valley walls
(Bentley et al., 2006; Stone et al., 2003; Sugden et al., 2005). Clasts sourced from rockfall or reworked from older ice marginal sediments in these regions of cold-based ice will not be sufficiently eroded to remove inherited nuclides. Ferrar Dolerite may be particularly susceptible to cold-based, non-erosive glacial transport because many high-elevation cliff faces are exposed over thin alpine and valley glaciers that are tributaries to larger outlet glaciers. Dolerite clasts sourced from these areas likely carry inherited nuclides that are not removed by glacial erosion.

When rockfalls from cliff faces deliver granite or dolerite onto glacier surfaces below, inherited nuclides remain in some clasts. While rockfall onto glaciers may shatter large volumes of rock and expose fresh surfaces some clasts may retain the original surface exposed on the cliff face. Clasts that land directly onto the ice surface will accumulate additional nuclides if they remain exposed during supraglacial transport. Rockfall that does not land directly on the glacier surface, but accumulates along outlet glacier margins, could be subjected to additional exposure prior to a glacial high stand that entrains
these clasts.

### 5.3.3 Ice sheet reworking of local volcanic rocks causes and prevents prior exposure

$^3He_{olv}$ exposure ages of basalt clasts, sourced from the local McMurdo Volcanic Suite, record local landscape exposure prior to and following ice sheet advance (Brook et al., 1995). The bimodal distribution of $^3He_{olv}$ in basalt exposure ages predates maximum ice sheet extent in McMurdo Sound (25.0 ka) and post-dates ice thinning (9.6 ka). Clasts with
inherited nuclides and excessively old exposure ages may have been initially sub-aerially exposed during a glacial low stand prior to the local LGM or the last interglacial (MIS 5). As ice advanced into McMurdo Sound during the local LGM, volcanic clasts with inherited nuclides may have been reworked into Ross Sea drift. However, clasts that lack previous exposure could be entrained from submarine sources that lack any previous exposure and transported along the base of the ice sheet in McMurdo Sound. As ice thinned and down-wasted, basal clasts would be exposed last, possibly producing
exposure ages younger than the onset of ice thinning (12.3 ka), a process which could also explain some of the younger $^{10}Be_{qtz}$ exposure ages of granite in Ross Sea drift. The $^3He_{olv}$ exposure ages are compatible with ice thinning on eastern Mount Discovery from its maximum extent after 14.0 ka to near present elevations at 7.3 ka (Anderson et al., 2017), as well as other outlet glacier thinning histories of that show rapid ice lowering during the Early Holocene (Anderson et al., 2020; Goehring et al., 2019; Jones et al., 2015, 2021; Spector et al., 2017) as grounded ice in the Ross Sea retreated (Halberstadt et
al., 2016).

### 5.4 Upper Discovery deposit: maximum glaciation during MIS 8

The Upper Discovery deposit marks the maximum limit of glaciation of McMurdo Sound during the end of MIS 8 (243 – 300 ka). The decreasing elevation of the Upper Discovery limit around Mount Discovery from 776 m in the east to



~450 m in the west suggests a larger ice sheet in the Ross Sea overflowed into McMurdo Sound during a glacial period prior

to the local LGM. As there are no erratic lithologies present above this limit, the Upper Discovery deposit delineates the largest and thickest ice sheet in the western Ross Sea to inundate McMurdo Sound since Mount Discovery formed 5.5 to 4.5 Ma (Kyle, 1990). We assign an age of late MIS 8 to the Upper Discovery deposit with consideration of exposure age scatter and nuclide inheritance. At the local LGM limit on Mount Discovery, Ross Sea drift $^3He_{pyx}$ ages exceed the $^{10}Be_{qtz}$ ages by 14.1 to 32.1 ka. If a similar magnitude of nuclide inheritance is assumed for the Upper Discovery deposit, three $^3He_{pyx}$

exposure ages in Ferrar Dolerite from the Upper Discovery deposit still date within MIS 8. One older $^3He_{pyx}$ dolerite exposure age from this deposit (ACX-13-61: 366.2 ± 11.6 ka) is compatible with an age of MIS 8; as observed with in Ross Sea drift, outlier $^3He_{pyx}$ exposure ages can be preserved in the same deposit. The $^{10}Be_{qtz}$ exposure age (ACX-13-52: 153.3 ± 2.0 ka, MIS 6), is too young. However, this sample was collected above a cavernous weathering pit on a granite boulder, which clearly indicates that the boulder was eroding, and thus losing some of the inventory of $^{10}Be$ that accumulated since

deposition; this would yield a young-biased exposure age.

        Maximum ice sheet expansion during the end of MIS 8 is compatible with other Pleistocene glacial records in McMurdo Sound. Many of the exposure ages from pre-LGM glacial sediments at lower elevations in McMurdo Sound are either younger or older than those from the Upper Discovery deposit; this suggests that older Pleistocene ice sheets in McMurdo Sound were thicker than during the local LGM but were less extensive than during MIS 8. The AND-1B core in

McMurdo Sound records ~5 glaciations during the last 500 ka (McKay et al., 2012). While poor core recovery in the upper part of AND-1B limits the interpretation of ice sheet behaviour during the last glaciation and interglacial, deformed till recovered from the third-to-last glacial period could reflect greater ice thickness in McMurdo Sound during MIS 8 (McKay et al., 2012).

        There is little evidence for a MIS 8 glacial high stand preserved in sediments along the margins of outlet glaciers

that flow into McMurdo Sound during glacial periods. Clasts from the head of the Skelton Glacier drainage basin likely record multiple episodes of exposure during interglacials and burial below ice during glacial periods from the Pliocene to Pleistocene, which is compatible with increased ice thickness during MIS 8 (Jones et al., 2017). At lower elevations along Skelton Glacier, most samples record ice-thinning following the LGM (Anderson et al., 2020). Due to ice cover, no cosmogenic nuclide studies from Mullock Glacier exist. Although Hatherton and Darwin glaciers transport less material into

McMurdo Sound during glacial periods (Talarico et al., 2013), there is also no evidence of MIS 8 glacial sediments preserved along the margins of those glaciers (Joy et al., 2014; King et al., 2020; Storey et al., 2010). In general, exposure ages of pre-LGM glacial sediments above outlet glacier margins record glacial high-stands during MIS 6 and earlier, but there is no apparent evidence for elevated outlet glaciers during MIS 8 (Anderson et al., 2020; Bromley et al., 2010; Hillenbrand et al., 2014; Joy et al., 2014; Staiger et al., 2006; Storey et al., 2010).

At a wider scale, maximum ice sheet extent in McMurdo Sound during MIS 8 may reflect the unique character of this glacial period relative to others of the past 800 kyr. During MIS 8, global ice volume was lower and average eustatic sea level was higher than the LGM (Spratt and Lisiecki, 2016); yet the marine-based ice sheet in McMurdo Sound was larger



than its local LGM configuration. In the EPICA Dome C ice core, the glacial termination following MIS 8 (T3) is characterized by an earlier warming between 256-249 ka that coincided with an insolation minimum in the northern

hemisphere (unlike other terminations of the past 500 kyr), followed by brief cooling between 248-249 ka, and then rapid warming until 243 ka (Bréant et al., 2019). The subsequent MIS 7e interglacial, while warm in Antarctica, was globally cooler than other interglacials of the past 450 kyr (Past Interglacials Working Group of PAGES, 2016). It is possible that warmer Antarctic temperatures increased accumulation, and reduced ocean forcing supported a thicker marine ice sheet in the western Ross Sea at the end of MIS 8.

**6. Conclusion**

Exposure ages of glacial deposits in McMurdo Sound record multiple surface processes operating in the Transantarctic Mountains prior to, during, and following glacial entrainment throughout the Pleistocene. Ross Sea drift contains clasts that yield exposure ages during the early Holocene and last glacial period, but generally exceed the timing of local LGM ice extent (12.3 – 19.6 ka) established by an independent radiocarbon chronology. We attribute the scatter and prior exposure in

this dataset to surface processes related to bedrock sources for glacially eroded and transported material. Granite clasts, sourced via sub-glacial plucking and/or rock fall, produce scattered exposure ages that are less likely to record prior exposure. Ferrar Dolerite clasts, sourced from exposed cliff faces and high elevation peaks in the Transantarctic Mountains above outlet glaciers, are more likely to contain inherited nuclides, due to entrainment along cold-based glacier margins. Local volcanic rocks in McMurdo Sound produce exposure ages that record landscape exposure prior to ice sheet advance

and following ice sheet thinning, which reflects reworking of previously exposed clasts along with clasts derived from sub-marine sources that lack any previous exposure. $^3He_{pyx}$ exposure ages of higher elevation, weathered glacial deposits on Mount Discovery suggest that the largest Pleistocene ice sheet advanced into McMurdo Sound during the latter half of MIS 8. Although prior exposure and boulder erosion limits a precise age of such older deposits, it may be rare geologic evidence of Antarctic ice sheet volume during MIS 8, a glacial period marked by generally warmer Antarctic temperature and higher

global sea level.

**Data availability statement**

All data presented in this manuscript are hosted in a repository at the Open Science Framework (Christ et al., 2021b).



## Author contribution

AC designed the study, conducted field work, and analysed data, and wrote the manuscript with contributions from all co-
authors. PB edited the manuscript and provided mentorship. JL, JS, and GW conducted laboratory analyses and edited the
manuscript.

## Competing interests

The authors declare that they have no conflict of interest.

## Acknowledgements

This project was funded by NSF Office of Polar Programs award 1246316. We appreciate the field assistance from E.
Chamberlain, N. Robinson, D. Rybarcyzk, S. Mackay, D. Kowalewski, and A. Canty; laboratory assistance from R. Schwartz
and J. Sparks.

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
