# Peer review of "Cosmogenic nuclide exposure age scatter in McMurdo Sound, Antarctica records Pleistocene glacial history and processes"

_Geochronology, 2021_

## Author Response (AR1)

**Author's response to reviews:**

**Cosmogenic nuclide exposure age scatter in McMurdo Sound, Antarctica records Pleistocene glacial history and processes**

Andrew J. Christ1, Paul R. Bierman1,2, Jennifer L. Lamp3, Joerg M. Schaefer3, Gisela Winckler3

1Gund Institute for Environment, University of Vermont, Burlington, VT, 05405, USA

2Rubenstein School of the Environment and Natural Resources, University of Vermont, Burlington, VT, 05405, USA

3Lamont Doherty Earth Observatory, Columbia University, Palisades, NY 10964, USA

**Responses to comments are show in *italic font*. Changes made in the text are shown in *red italic font*. Line numbers all refer to the clean, non-track changes version of the revised manuscript.**

**Response to editor comments:**

Dear Authors,

Thanks for submitting the revised manuscript along with the detailed responses to the critical comments suggested by two expert reviewers.

As you know, the two reviewers provided two important comments such as nucleogenic (or inherited) 3He production and Interpretation of MIS-8 Ross Sea Drift.

I found the first concern is now clearly corrected and explained in the revision. The second concern was well defended but needs more explanation in the revised manuscript.

Specifically, any influence on the interpretation of the apparent ages by "surface erosion (actually weathering) rate of the sampled boulder surface. Definitely it should yield older ages. I know the correction for erosion rate is not straightforward but the readers would expect at least the possible explanation.

**Response:** We have addressed this in the discussion section about the Upper Discovery deposit. Specifically, we explain that the particular granite boulder that was sampled had a cavernous weathering pit indicating that this sample was weathering leading to a loss of the cosmogenic 10Be inventory. When a Antarctica specific erosion rate of 0.13 mm/kyr is applied to this granite sample, the erosion-corrected age is (147 ka). When a dolerite specific erosion rate is applied (0.19 mm/kyr), the dolerite exposure ages of the same deposit are still much older (226 – 356 ka). We are hesitant to calibrate the 10Be exposure age of this single boulder based on an erosion rate that is apparently much higher than the regional average.

In the clean version of the revised manuscript we have added the following (lines 404-410):

"Even if lithology-specific boulder erosion rates for granite (13 mm kyr-1) and dolerite (19 mm kyr-1) (Marrero et al., 2018) in Antarctica are considered, the difference between the exposure ages of the granite boulder (147 ka) and the dolerite boulders (~226-356 ka) is too much to be explained by the difference in erosion rates between lithologies. This suggests that this particular granite boulder was weathering much faster than regional average rates or it may have been affected by post-depositional movement. Additional exposure age constraints from the Upper Discovery deposit as well as other higher elevation Pre-LGM glacial sediments in the McMurdo Sound region will further test if a glacial high-stand occurred during MIS 8."

Other minor comments and technical corrections are well modified.

I would add some minor comments/corrections below, which I would like you to incorporate the final version of the manuscript.

- Fig. 2, "Trans Antarctic Mountains": Could you upside down it just like other characters?

Response: Fixed as you suggested

- The revision still does not provide "production rate of 3He and 10Be". Which number did you use, global or local (Antarctic)? Please provide specific number (e.g. 4.3 atoms/gyr).

**Response:** We added these details in the methods section: global 3He production rate  $(120 \pm 9.4 \text{ atom } g^{-1} \text{ yr}^{-1})$  (Goehring et al., 2010) [line 185] and global 10Be production rate (4.15 atoms  $g^{-1} \text{ yr}^{-1})$  (Martin et al., 2017) [line 198].

- Local LGM: The present study suggests a local LGM occurred at 19.6-12.3 ka, based on 14 ages of fossil algae embedded in Ross Sea Drift. I think the present conclusion assumes that the dated RSD is the terminal or latero-frontal moraine which marks the most extensive position of the glacier system. Considering the extensive characteristic of glacier ice during LGM (simply glaciation peak), the ice may have been a unified body particularly in the low elevations where the RSD is found.

Is there any possibility that the RSD can be a recessional or retarded lateral moraine? It is interesting that the local LGM at Terra Nova Bay (only ~100 km north) occurred during "MIS 4", not during MIS 2 (Rhee et al., QSR, 2019. Timing of the local last glacial maximum in Terra Nova Bay, Antarctica defined by cosmogenic dating).

**Response:** The timing of the local LGM in McMurdo Sound is based on 237 calibrated radiocarbon ages from the maximum limit of Ross Sea drift on the volcanic islands and peninsulas of McMurdo Sound (Christ & Bierman, 2020) as well as headland moraines in the ice-free valleys of the Royal Society Range (Hall et al., 2015; Jackson et al., 2018; Hall et al., 2001). The lower elevations of Ross Sea drift could indeed be part of a recessional moraine deposited as the grounded ice sheet thinned and retreated. In this manuscript we are only examining exposure age samples that were collected from the maximum limit of the ice sheet, so we think that our interpretations remain consistent with the glacial geomorphology of the area. The timing of the local LGM from radiocarbon ages of glacio-fluvial and glaciolucustrine deposits along the former ice margin agree well with glacial marine sediment age constraints that show the transition from grounded ice to either a floating ice shelf or seasonally open marine conditions after 8 ka just north of Ross Island.

Thanks for directing me towards Rhee's 2019 paper.

I recommend "Publish after some minor revisions".

Associate Editor, Yeong Bae Seong.

Thanks so much Dr. Seong. You may not remember this, but I collected my very first cosmogenic nuclide sample with you on a small ice-free bedrock surface on the Danco Coast in the Antarctic Peninsula while we were aboard the ARAON in spring of 2013. Thanks for being my first teacher in cosmogenic nuclide field methods nearly a decade ago!

**AUTHOR RESPONSE TO REVIEWER COMMENT #1**

**Cosmogenic nuclide exposure age scatter in McMurdo Sound, Antarctica records Pleistocene glacial history and processes**

Andrew J. Christ, Paul R. Bierman, Jennifer L. Lamp, Joerg M. Schaefer, and Gisela Winckler

**Author responses are recorded in italics below reviewer comments.**

**General Comments:**

In this paper, Christ et al. present a new surface-exposure dataset from the McMurdo Sound region of the Ross Sea in Antarctica. Although the high prevalence of inherited cosmogenic nuclides in local sediments makes surface-exposure dating in the region a challenge, here the authors use a nearby radiocarbon chronology to benchmark their data and to enable direct comparison of their apparent exposure ages with the timing of the local Last Glacial Maximum (LGM). They also recalculate exposure ages from previously published studies to enable a synoptic view of regional exposure ages and inheritance.

Their results indicate that although inheritance is indeed pervasive in the sampled glacial sediments, the ultimate pattern of exposure-age scatter is in part dictated by lithology and associated transport history. For example, clasts derived from subglacial sources appear to best reflect the timing of local deglaciation. In contrast, clasts sourced from areas above glacial trimlines produce exposure ages suggestive of possible nuclide inheritance. Following this analysis, Christ et al. assess potential longer-term (pre-LGM) patterns of glaciation in McMurdo Sound. They suggest that the pre-LGM Discovery drift unit was deposited during MIS8, highlighting the utility of surface-exposure dating to investigate surface processes and landscape evolution through time.

This paper illustrates an excellent application of larger exposure-age datasets. It is well written and well presented, and I appreciate their thoughtful discussion of how different sediment sources with unique histories may impact surface-exposure chronologies. There are a few areas where I think the authors need to add additional detail or justification for their methods and interpretations. I also have one larger comment centred on their discussion of 3He ages from local dolerite. I detail these comments below, as well as a few technical comments/corrections.

**Response to general comments**: Thank you very much for your well-stated summary of work and thoughtful comments. We will address your general comments about our methods and interpretations, and specific comments about  ${}^{3}\text{He}_{pyroxene}$  ages below.

**Specific Comments:**

The authors note that Ross Sea drift pyroxene 3He ages predate nearby quartz 10Be ages by 14-32 kyr (line 384). They assign this offset to differing mechanisms of clast transport and deposition and suggest that the age offset may be explained by 'inherited' cosmogenic 3He. Previous studies show that Ferrar Dolerite pyroxenes contain non-zero amounts of non-cosmogenic 3He (see Ackert, 2000; Margerison et al., 2005; Kaplan et al., 2017; Balter-Kennedy et al., 2020). This amount is generally around 5-7 x 10^6 at/g, and so significant over the timescales of interest here. In particular, this amount could account for some or all of the apparent offset between the Ross Sea drift 10Be and 3He ages. This could lessen the need for a depositional mechanism in this case. If the authors have a 'shielded' piece of Ferrar Dolerite on hand from their field site they could measure this non-cosmogenic amount directly. Alternatively, they could assume a non-cosmogenic 3He is not present within their samples they should make that clear within their discussion and interpretations. In any case, I would encourage the authors to discuss this point within their text.

**Response:** Thank you for pointing out the need for a nucleogenic  ${}^{3}He_{pyx}$  correction. In the submitted version of the manuscript we did not apply this correction. Unfortunately, we did not collect "shielded" Ferrar Dolerite samples to directly measure the non-cosmogenic  ${}^{3}He$  contribution at our field site. We recalculated the dolerite exposure

ages using the correction of 3.3E+06 atoms/g reported by Balter-Kennedy et al. (2020), as well as the 5E6 to 7E6 at/g correction that you suggest. These corrections decrease the  ${}^{3}He_{pyx}$  exposure ages of dolerite samples in Ross Sea drift by ~12.6 kyr (3.3E6 at/g correction), 19 kyr (5E6 at/g correction), and ~26 kyr (7E6 at/g correction). See the table below for a comparison of the non-correct and corrected ages (using the LSDn scaling scheme) below. Regardless of the correction, nearly all of the exposure ages of dolerite in Ross Sea drift are older than the timing of the local LGM, indicating that our original observation about inherited nuclide inventories in dolerite clasts remains valid. This sensitivity test suggests that the 7E6 at/g correction is likely too much for these samples, as it produces an apparent exposure age that appears modern (ACX-13-08: 161 yrs). This would be the only sample in the entire dataset (regardless of lithology or nuclide) to generate such a young age. The 5E6 at/g correction produces an apparent exposure age (14.3 ka) that corresponds to the timing of the local LGM in McMurdo Sound. In the manuscript. We will report exposure ages using the 3.3E6 at/g nucleogenic correction reported by Balter-Kennedy et al., 2020 as this is the most up-to-date value used in the Antarctic cosmogenic nuclide community and produces exposure ages that are more plausible than higher correction values.

No 7E6 at/g nucleogenic 3.3E6 at/g nucleogenic 5E6 at/g nucleogenic Sample name nucleogenic correction correction correction correction Difference Difference Difference Age (yr) Age (yr) Age (yr) Age (yr) 26,978 14,300 161 ACX 13 008 -12,678 7,752 -19,226 -26,817 ACX\_13\_009 44,503 31,938 -12,565 25,407 -19,096 17,722 -26,781 ACX\_13\_012 42,320 29,565 -12,755 23,014 -19,306 15,307 -27,013 ACX\_13\_048 255,752 245,901 -9,851 240,827 -14,925 234,857 -20,895 239,440 229,235 -10,205 224,133 -15,307 218,130 -21,310 ACX\_13\_052 347,931 335,043 ACX 13 061 357,598 -9,667 341,487 -16,111 -22,555 ACX\_13\_068 242,780 232,358 -10,422 226,990 -15,790 220,674 -22,106 376,717 -15,089 344,709 ACX\_14\_005 361,628 353,854 -22,863 -32,008 -24,791 ACX\_14\_015 52,310 35,999 -16,311 27,519 17,543 -34,767

In the revised manuscript we will include the information about the  ${}^{3}He_{pyx}$  nucleogenic correction in the methods section and cite the papers (Ackert, 2000; Balter-Kennedy et al., 2020; Kaplan et al., 2017) you have kindly supplied.

Changes in text: Line 179-181: We subtracted a non-nucleogenic 3He correction of 3.3 x 106 atoms g-1 (Balter-Kennedy et al., 2020) to all 3He measurements, as this is the most up-to-date correction measurement and higher correction values (5 - 7 x 106 atoms g-1) generate some exposure ages that produce modern exposure ages that are unreasonably young.

The authors use the "LSDn" scaling scheme for their exposure age calculations. While I see no problem with this they should include a few lines to justify this choice. Why is "LSDn" preferable for this location or time period versus an alternative scheme? If the authors chose an alternative scheme, would their interpretations change? For example, would samples still fall within the proposed MIS8 window using an alternative scaling scheme such as "St"? Or would younger samples still correlate with radiocarbon ages of Ross Sea drift? As the "LSDn" scheme can produce higher production rates relative to alternative schemes such as "St" or "Lm", justifying their choice of scheme here is key.

**Response:** Thanks for bringing attention to this, we recognize that we should have clarified our decision about the scaling scheme. We employed the LSDn scaling scheme, which is time dependent, because the compiled dataset spans a wide timescale over the past 500 kyr. As you have suggested, we applied the LSDn, Lm, and St scaling schemes for sensitivity testing on the exposure age dataset. Regardless of the scaling scheme applied, we still observe the same trends according to nuclide and lithology. The LSDn scheme indeed produces younger exposure

ages than St or Lm, but the difference is usually less than 1 kyr for samples with exposure ages <50 ka. None of the samples younger than 20 ka in McMurdo Sound have differences greater than 880 yr; this means our interpretations about the exposure age scatter relative to the radiocarbon constrained timing of the local LGM are not affected. The exposure age difference between scaling schemes becomes greater for older samples, but again does not affect our interpretations even for samples from Mount Discovery that correspond to MIS 8. As we revise the paper, we will include these details about the scaling scheme sensitivity testing.

Change in text: Line 184-189: "We employed the LSDn scaling scheme, which is time dependent, because the compiled dataset spans a wide timescale over the past 500 kyr. We note that exposure ages using the St or Lm scaling schemes generate slightly older exposure ages, but do not change observed patterns in the wider dataset."

Related to the above, although the authors note their chosen scaling scheme I was unable to find any discussion of the nuclide production rates used for exposure age calculations. As they use the online calculator [hess.ess.washington.edu] I assume this means that they utilise the standard/default 'global' production rates provided, but this should be clarified.

**Response:** You are correct that we mistakenly omitted explaining the production rate used in our calculations. We used the global production rates supplied by the online calculator. We will clarify this in the revised manuscript.

In addition, what atmospheric model is used for exposure-age calculation? I presume they used the Antarctic 'ANT' standard of Stone (2000), but it is best to list all calculation parameters to ensure reproducibility.

**Response:** Yes – thanks for calling attention to this. We did use the ANT standard of Stone 2000. We will include this detail in the revised manuscript.

Change in text line 181-185 relevant to the two comments above: "3He in pyroxene ( ${}^{3}He_{pyx}$ ) exposure ages were calculated using Version 3 of the online exposure age calculator hosted by the University of Washington (https://hess.ess.washington.edu/) (Balco et al., 2008) assuming density of 2.9 g cm-3, corrected for shielding and thickness, the Antarctica-specific atmospheric model (ANT) (Stone, 2000), the global 3He production rate (120 ± 9.4 atom g-1 yr-1) (Goehring et al., 2010), and used the LSDn scaling scheme (Lifton et al., 2014)."

Figure 5 is an excellent visual synopsis of the data, but would it be possible to indicate which samples come from each lithology? Perhaps using additional colours or shapes? As lithology is such a central component of the overall discussion I think including this element would be very useful for the reader.

**Response:** Great suggestion – we will change the symbology to different shapes to show different lithologies. We revised Figure 5 to show target nuclide and lithology as different symbol shapes and used different colored symbols for Ross Sea drift (yellow), pre-LGM deposits (orange), and the Upper Discovery deposit (purple). See new figure below.